# A Study of the Implications of the European Securitisation Regulation 2017/2402 on Malta

**Joseph Micallef [1], Simon Grima [2,*], Sharon Seychell [2], Ramona Rupeika-Apoga [3] and Mark Lawrence Zammit [2]**

1    Faculty of Law, University of Malta, Msida MSD 2080, Malta; joseph.micallef.10@um.edu.mt
2    Faculty of Economics, Management and Accountancy, University of Malta, Msida MSD 2080, Malta;
     sseyc01@um.edu.mt (S.S.); mark.l.zammit.15@um.edu.mt (M.L.Z.)
3    Faculty of Business, Management and Economics, University of Latvia, LV-1586 Riga, Latvia; rr@edu.lu.lv
*    Correspondence: simon.grima@um.edu.mt

**Abstract:** A decade ago, the financial world was taken by surprise, when prominent credit institutions filed for bankruptcy. The financial crisis phenomena spurred the need for regulating Securitisation and enhancing the capital requirements framework. In response, the Basel Committee initiated the regulatory treatment for the Simple Transparent and Comparable Securitisation (STC Securitisation), the USA passed the Dodd–Frank Act and the EU introduced Securitisation Regulation No. 2017/2402 to address the causes and failures, which were identified, following the aftermath of this financial crisis. With this article, we aim to analyse the main provisions of the Regulation No. 2017/2402 on Malta as a jurisdiction for securitisation and provide an insight on the prospective market development. To reach our aim we analysed scholarly documentation (academic chapters, journals, articles and monographs), rules, guidelines, recommendations, directives and regulations and use the case study methodology, as suggested by Yin (2003) and Yazan (2015), on Malta. In our opinion, recently, Malta has made significant improvements in the securitisation sector, mostly evidenced by the introduction of the legislation. All interviewees emphasised that Malta has substantial opportunities for further growth in the securitisation market and it is encouraged to be exploited well.

**Keywords:** Simple Transparent and Standard Securitisation (STS Securitisation); Capital Markets Union (CMU); Capital Requirement Regulation (CRR); Basel Securitisation Framework; net economic interest

## 1. Introduction

Securitisation is a structured process whereby homogenous financial assets are pooled, underwritten, and sold to outside investors in the form of securities (Sarkisyan and Casu 2013). The securitisation phenomenon originated in the United States of America (US), in the early 1970s, primarily with the structuring of mortgages loans by the US government agencies. Distinctively, securitization started developing in the European Union (EU), due to the demand from an institutional investor; the importance of financial innovation and later the introduction of the single currency.

Throughout the years, securitisation has shaped the credit institutions' operations drastically, leading to a new source of financing of investment opportunities. Credit institutions moved away from the traditional view of deposit institutions to a more hybrid approach as purely intermediaries between borrowers and the capital markets, which shifted from the traditional "originate-to-hold" model to a more modernised "originate to distribute" model. This model has enabled credit institutions to collateralize assets for the issue of the short-term paper, whilst integrating this technique to manage its liquidity risk practices. Asset securitisation attracted different market players which were less

regulated. The so-called "shadow banking system" functioned the same way as traditional credit institutions, however with minimal regulation and without access to the central bank funding.

Without a doubt, the unexpected outbreak of the 2007–2009 financial crisis, severely impacted the financial industry. Securitisation was implicated in the financial turmoil following the collapse of Lehman Brothers. Collectively, market participants suffered from the actions of other market participants, respectively, leading to a significant amplification of systemic risk within the financial markets. According to the European Commission, in 2014 the volume of securitisation in the EU had dropped by 42% compared to the average levels in the pre-crisis period (2001–2008). The commission's findings show that the amount of Small and Medium-sized Entities (SME) loan securitisation dropped from EUR77 billion in 2007 to EUR 36 billion in 2014. The commission estimates show that if the volume of EU securitisation reached its pre-crisis average, it would generate between EUR 100–150 billion in additional funding (European Council 2018).

The Federal Reserve and the European Central Bank (ECB) instituted the first attempt for a sound economic recovery by implementing a series of quantitative easing measures. Since then, reforms have been on the policymakers' agenda to revive the securitisation market. While the regulators are fully aware of the potential benefit of the market, they are aiming for a more transparent and compatible market. The US implemented the Dodd–Frank Act in 2010 as a direct consequence of the financial crisis of 2007–2009 with strict rules for disclosure, risk retention, and credit rating reforms (Grima 2012).

The European Union (EU) amended some directives (Directives 2009/65/EC, 2009/138/EC and 2011/61/EU) and regulations (Regulations (EC) No 1060/2009 and (EU) No 648/2012) indirectly related to securitisation. However, the securitisation market was still under-regulated. Strict regulatory requirements were introduced, i.e., Regulation (EU) 2017/2402 of the European Parliament and of the Council of 12 December 2017 laying down a general framework for securitisation and creating a specific framework for simple, transparent and standardised securitisation. The regulation classifies structured investment products, as to whether it satisfies the Simple, Transparent Securitisation criteria (STS criteria) according to an established criteria. National Competent Authorities (NCAs) are empowered to issue sanctions and warnings, if rules are not adhered to.

## 1.1. Significance and Originality of the Study

Several authors such as Schwarcz (2008, 2011), Cullen (2017), Bavoso (2013), highlight essential themes related to the future of Securitisation in Europe. Indeed, a regulatory assessment results in a clear picture of the state of the market and any potential weakness are identified. Although, there are specific studies, which relate directly or indirectly to the Securitisation legislation (Galea 2016), the role of Special Purpose Vehicles (SPV)'s in the market (Zammit 2010), on the reinsurance perspective (Buhagiar 2015) and EU regulations such as Undertakings for the Collective Investment in Transferable Securities (UCITS) (Bennetti 2012); Alternative Fund management Directive (AIFMD) (Meli 2014), Markets in Financial Instruments Directive (MiFID) (Grech 2017), we still believe that more has to be carried out to highlight the necessity and impact of this specific regulation and the implications on Malta as a securitisation jurisdiction.

It is our opinion that the newEU regulation is so far working efficiently, even though this regulation introduced compliance costs for both the National Competent Authorities (NCAs) and the counterparties, respectively. However, this is the road to recovering and building confidence in the securitisation market. On the other hand, the NCAs need to ensure that the necessary training is provided to the employees to carry out the required regulatory obligations, while at the same time, the importance of maintaining risk management principles by the counterparties would prove beneficial to the securitisation structure.

## 1.2. The Importance of Securitisation

Even though securitisation created hurdles in the financial market, it is essential to highlight the advantages, which bring about potential benefits to the issuers, investors and the economic system.

Securitisation is a unique financial tool because the benefits can be achieved with different objectives and different market characteristics.

Due to the nature of their operations, credit institutions face asset-liability mismatching[1], which occurs when the assets and liabilities do not correspond to their maturity term. Hence, the tranching of securities ensure that the Asset Backed Securities (ABS) issued, meet the respective investor risk profile. Another benefit is the diversification exposures, which offer flexibility whereby long-term investors are willing to buy long-term assets given the liabilities duration.

Securitisation benefits are sought by non-credit institutions, in order to diversify their lending base. These institutions are less vulnerable when directly compared to credit institutions, especially during a recession period. Non-credit institutions are less prone to balance sheet risks since they are less correlated, have less leverage and have a less complicated balance sheet (Deloitte 2018) and (Grima 2012).

Another significant use of securitisation is the high-quality collateral as securitisation transforms highly illiquid loans into liquid assets. The EU Regulation 2017/2402[2] is beneficial for the originators to increase the collateral level (European Commission 2020) and (Global Legal Book, Securitisation 2018).

Although traditional securitisation is still popular in the financial world, synthetic securitisation is gaining momentum. Synthetic securitisation is distinctively different from the traditional securitisation given that no true sale of assets occurs. An advantage of synthetic securitisation over the traditional securitisation, is that it offers flexibility to the underlying loan portfolio and is cost-effective both from a legal and operational perspective (Deloitte 2018); (Messina and Horrocks 2019) and (Grima 2012).

The development of the securitisation regulation unlocks credit supply and leads to the desired economic recovery, especially concerning the European Small and Medium-sized Entities (SMEs). These entities are still reliant on credit institutions and have not fully exploited the market either directly by accessing the securitisation market or indirectly through the development of other securitisation segments that free up space on bank balance sheets, which increases their lending power. Financing of the SME's working capital has proved to be beneficial whereby these entities have access to additional capital to finance their assets (Orkun 2013); (Deloitte 2018); (Baums 1994) and (European Commission 2020).

## 2. Research Objectives

In order to assess the implications of the securitisation regulation, we have carried out an analysis on the main provisions of the Regulation No. 2017/2402. Consequently, various proposals at EU level and other studies conducted during the implementation phases were consulted. This was done so as to (1) Improve the understanding of the securitisation and facilitate deeper knowledge of its structure and functioning; (2) Analyse the importance of the STS criteria for securitisation and (3) Gather factual evidence from industry experts on how the regulation impacts Malta as a small state jurisdiction.

### 2.1. Research Questions

During the implementation phase of this regulation, the European Commission conducted an impact assessment on possible implications. The different market practices, which existed in the Member States and the period in the timeline of the regulation, when the impact assessments were carried out, make it difficult to formulate a clear opinion and understanding and hence the following research question have been raised:

What is the benefit that securitisation can add locally (in Malta)?

What type of requirements and costs shall the securitisation regulation bring about?

What will be the implications on Malta's securitisation industry?

---

[1]　An asset-liability mismatch occurs when the assets and liabilities do not correspond to their maturity term.

[2]　REGULATION (EU) 2017/2402 OF THE EUROPEAN PARLIAMENT AND OF THE COUNCIL of 12 December 2017.

### 2.2. Why Malta?

Malta, being a small island state (about 316 km$^2$) and a population of circa 450,000 EU citizens, is one of the six small states within the EU and Eurozone (population of under 3 million). Moreover, the financial services sector is a major pillar of Malta's economy attributed mainly to the advantageous tax regime, a low-cost environment, a well-trained and motivated English-speaking labour force and an EU-compliant, yet flexible, domicile. The specific features about Malta's financial system and history make this contribution towards this study particularly interesting for other small and larger countries/states within the EU and commonwealth countries Bezzina et al. (2014). Ultimately, the use of islands as small-scale laboratories for more complex politics, regulations and policies of larger countries has been highlighted by various prominent researchers (Bezzina et al. 2014; Briguglio 1995; King 1993).

## 3. Post-Financial Crisis Regulation

### 3.1. Basel III: Strengthening the Risk-Based Capital Framework Following the Financial Crisis

The latest financial crisis revealed several weaknesses in the way banks managed their risks, due to a significant level of leverage involved. Several instruments being traded were exposed to credit risk, which also lacked price transparency and market liquidity. Credit institutions failed to undertake the appropriate risk assessment in time. In June 2011, the Basel Committee introduced the Basel III standard to strengthen the risk-based capital framework of the previous standard. The main shortcomings identified and addressed by the committee were:

1. Mechanistic reliance on external ratings;
2. Excessively low-risk weights for highly rated securitisation exposure;
3. Excessively high-risk weights for low rated senior securitisation exposure;
4. Cliff effects;
5. Insufficient risk sensitivity of the framework.

(Basel Committee on Banking Supervision 2016)

Indeed, the securitisation regulation highlights the importance of risk sensitivity, prudence, and consistency in credit risk. Risk management was given the utmost importance to set appropriate capital charges suitable for the risks undertaken. Another essential objective included the ability of comparison between banks, transparency and the simplicity of securitisation.

Although, the Internal Rating Based (IRB) approach was introduced in Basel II, the committee decided to retain the IRB approach in Basel III. However, higher asset value correlation parameters were introduced. The financial crisis effect led Basel to increase the specific type of exposures. Consequently, the capital requirements were enhanced by the Basel Committee and exposures of US$100 billion dollars by financial institutions are now subject to 25% higher asset value correlation. Nevertheless, the shadow banking effect led the committee to increase asset correlation exposures with the concerned unregulated market participants.

Basel III requires banks to hold higher levels of Common Equity Tier I. An increase in minimum Common Equity Tier (CET) 1 Risk Based Capital Ratio (RBCR) has been raised from 2% to 4.5% and the Minimum Tier 1 RBCR from 4% to 6% with adoption date established as of January 2019 (Schwarcz 2011). Tier III capital was eliminated, to prevent the inflation on the capital base. The issue of dividends took prominence in the recent financial crisis, which was tackled by the committee by restricting the earnings allowed to be distributed as dividends.

Statistical methods were introduced to track the bank's chaining exposure to insolvency risk over time. No additional improvements were evident when compared to the previous standard. However, an exception was made only to the credit risk in the Basel II standard, whereby two new approaches were introduced, the Standardised Approach (SA) and the IRB approach. The main distinction between the two approaches used depends on the type of the credit institution. Sophisticated credit institutions

should apply the IRB, while the less sophisticated should apply the SA. The latter approach depends on the credit institution's exposure in a securitisation. While the IRB approach acknowledges a broader assessment of the relevant risks associated with the securitisation exposure (Grima 2012).

### 3.2. A New Hierarchical Concept for the Revised Securitisation Framework

A new hierarchical concept introduced in 2014 by the committee, aimed to make capital requirements more prudent and risk sensitive. This concept seeks further simplification and less reliance on external credit ratings.

Figure 1 illustrates the hierarchy of approaches in the revised framework for securitisation.

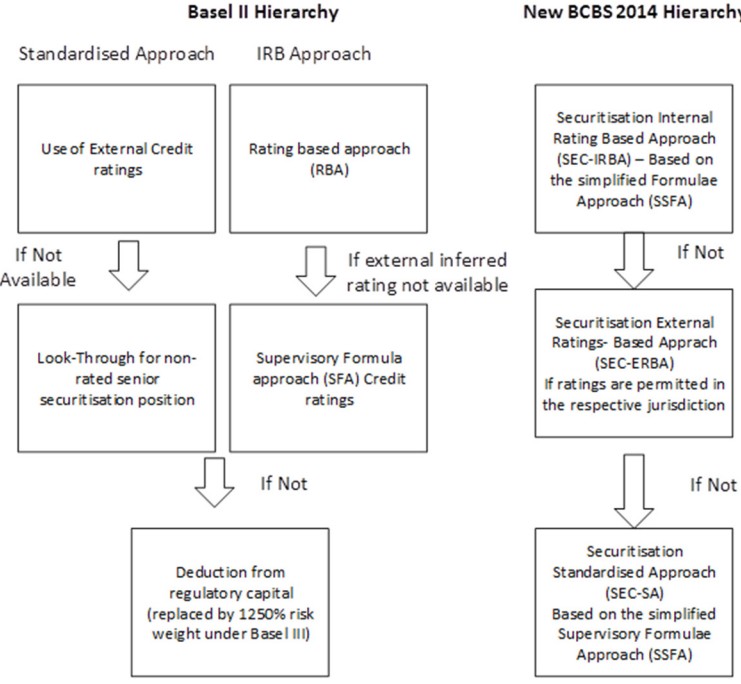

**Figure 1.** Hierarchy of approaches in the revised framework for securitisation. IRB: Internal Rating Based. Source: adapted from (Segoviano et al. 2015).

Basel Committee on Banking Supervision (BCBS) hierarchy introduces the Securities Exchange Commission (SEC)–IRB Approach at the top of the regime, followed by SEC–External Rating Based (ERB) Approach and SEC-Standardised Approach (SA) (Basel Committee on Banking Supervision 2016). This approach is based on the Simplified Supervisory Formula Approach (SSFA) (Segoviano et al. 2015). A vital input of the SSFA formula is denoted by $K_{IRB}$ (Internal rating-based Approach (IRB) capital requirements for the underlying pool of securitised assets), which is the capital requirements for the underlying exposure for the IRB framework. The capital charge includes the expected loss and in certain applicable cases, the dilution risk. Furthermore, if an SPV is involved in the securitisation structure, the separate exposures are to be included. The hierarchy provides the option to use a different approach if the previous one is not adequate for the given exposures. All three approaches illustrate a more risk-sensitive approach, given that the supervisory parameter (denoted by 'p') concerning to SEC–IRB Approach and SEC–SA. While a downward adjustment is about the risk weighting of the relevant risk tables issued by Basel. A bank is excluded from making use of any approach out of hierarchy, given the fact that an assigned risk weight of 1250% is set for a given securitisation exposure.

A significant shortcoming in the SSFA specifically in Basel II Hierarchy was the lack of adequate incorporation of maturity, which are captured partially in the KIRB. Several revisions were included in comparison with previous Basel standard. Basel III framework included a change of the previous assigned risk weights relating to external rating, the seniority and the granularity of the underlying

pool. Revisions in the external rating included additional risk drivers such as the tranche thickness of non-senior tranches and tranche maturity.

Another shortcoming of the Basel II Rating Based Approach (RBA) led the committee to address the tranche thickness issue in the Basel III standard. CRA(credit rating agencies) issue a rating, depending on the tranche thickness, especially to any mezzanine tranche involved. A risk characteristic considered in the Basel III standard is the tranche maturity, which indicates the remaining effective maturity. The rating agency typically targets a given level of expected loss per rating, while the capital charge reflects its expected loss rate conditional on the assumed stress event occurring (Schwarcz 2016). However, the unstressed expected loss rate is not enough to determine the stressed expected losses. Indeed, the capital charge should reflect the expected loss rate depending on the stress event considered by comparison for tranche maturity. The introduction of a floor for one year and a cap of five years were included by the committee.

### 3.3. The Introduction of the Simple Transparent Comparable (STC) Securitisation within the Basel Capital Framework

International reforms were on the Basel committee's agenda, including the revised securitisation framework and the introduction of the simple, transparent and comparable (STC) securitisation. The revised framework was published in December 2014 and with an effective implementation date in January 2018. One must also appreciate the joint effort involved whereby the Basel Committee and the International Organisation of Securities Commission (IOSCO) identified factors hindering the securitisation market. Both regulatory players recognised the importance of the STC criteria as a crucial principle for a securitisation structure. However, the Basel Committee was involved in consultations whereby comments were put forward by the industrial stakeholders. The STC criteria excluded the Asset-Backed Commercial Paper (ABCP) since it is considered to be short-term securitisation.

The new capital requirements complement the introduction of the Simple, Transparent Comparable (STC) criteria by the Basel Committee. Intentionally this criterion would assist transaction parties to better evaluate the risk-return in a securitisation transaction, with the ability to compare across another securitisation. This criterion assists investors to carry out their appropriate due diligence. Simplicity criteria ensure that homogenous underlying assets are not overly complicated, which would help investors and supervisors to assess the risk involved. The disclosure of the adequate information on the underlying assets assists investors in understanding the risks involved. Information should not hinder transparency but should support investors when carrying out their assessment. The comparability criteria assist the investors in understanding such investment and to compare across other securitisation structures within an asset class taking into account different jurisdiction rules.

Of particular importance, is the framework introduced in December 2014, which included some crucial factors such as credit pool quality, attachment and detachment points of tranches. The framework did not add any features related to qualitative securitisation. However, the STC criteria capture the qualitative factors and ensuring mitigation of uncertainty related risks involved. By implementing the STC criteria, the risk sensitivity would increase and might introduce significant operational burdens (Basel Committee on Banking Supervision 2016).

STC criteria are approached from a regulatory capital perspective. Both the Committee and IOSCO acknowledged that STC criteria are useful in setting the regulatory capital requirements, with particular reference to credit risk. Moreover, in July 2015, the Committee issued additional guidance for the STC criteria, which ensures distinction of different capital treatment between STC and another securitisation transaction. Three essential enhancements were introduced to the criteria, which are the following:

1. More explicit requirements for minimum performance history;
2. The exclusion of transactions if standardised risk weights for the underlying exposures exceed certain levels;
3. A more explicit definition of granularity

([Basel Committee on Banking Supervision 2016](#)).

Originators are obliged to provide securitisation, which is compliant with the STC criteria, for regulatory capital purposes. On the other hand, investors need to undertake their own assessment whether it qualifies as STC criteria. If inaccurate information is provided, the originators would be liable for any breaches. Regulatory capital treatment is to be reviewed by supervisors for the credit institutions under their supervision. Any lack of regulatory capital concerning the STC criteria, supervisors are encouraged to take remedial action either under the Pillar 2 framework or else by the denying the regulatory capital treatment.

## 4. The Capital Markets Union

*Strengthening the European Financial Economy: The Capital Markets Union Project*

As international agenda flourished with the increased regulatory approach, the EU was lagging behind for a long-term vision implementation. The European Commissioner, Dr. Jean Claude Juncker initiated the programme for strengthening the financial economy before the European Parliament elections in 2014. Once elected to power, the Capital Markets Union (CMU) agenda was a top priority, included in the Commissioner for Financial Services portfolio. The Green Paper entitled 'Building a Capital Markets Union' drafted in 2015, intended to complement other EU projects such as the Banking Union, the Economic Monetary Union and the Single Market. Although the Green Paper discusses the importance of developing the securitisation market in the EU, other markets and operational issues were reviewed. This project sought to enhance both monetary policy and promote financial stability within the EU. Ultimately, strengthening the financial system, with the result of having a broader and more liquid market ([European Central Bank 2015](#)).

Substantial investment packages proved insufficient to kick-start national economies, due to a lingering problem in the capital market development, which was still evident. The Commission's proposal for a successful action plan built on implementing the short-term objectives. However, the long-term CMU objective should be maintained with early interventions in specific areas for a better functional capital market. Issues related to harmonisation of insolvency law and financial products taxation are challenging but quite essential. Another significant development which the CMU seeks to attain is market efficiency, leading to resources being available for the right opportunity at the lowest cost possible.

The Green Paper acknowledged that European businesses remain heavily reliant on banks for funding and relatively less on capital markets. Over-reliance on banking services in the EU and the lack of alternative financing channels led to several problems on the bank lending model. The revival of the Securitisation market could be beneficial to the CMU, with new initiatives introduced and especially to the SME's. Moreover, international regulators have already launched and promoted the importance of STC securitisation. The STS securitisation is included both in the CMU proposal and in the securitisation regulation. The road ahead for a functioning CMU is quite challenging and requires on-going commitment, which would lead to smoothing consumption and investment while managing the economic risk.

## 5. The EU Securitisation Regulation No. 2017/2402

The lack of harmonisation concerning the directives and regulations was still quite evident amongst EU member states. In September 2015, the commission proposed a regulation to harmonise the securitisation market at EU level. Initial consultations were held with different key stakeholders such as the Basel Committee on Banking Supervision (BCBS), International Organization of Securities Commissions (IOSCO) and the European Banking Authority (EBA). The regulation was approved and enforced on the 28th December 2017. However, the implementation date was 1st January 2019. A supplement of the European delegated regulation, together with a significant number of

regulatory and implementing technical standards[3], seek to achieve the desired aims and objectives. These standards ensure further clarity and transparency for market participants, whilst fostering supervisory convergence and co-operation at EU level.

The fundamental objectives of the regulation are based on the lessons learned from the recent financial crisis, which include the following:

1. to revive markets on a more sustainable basis so that STS securitisation can act as an effective funding channel to the economy;
2. to allow for efficient and effective risk transfers to a broad set of institutional investors;
3. to allow securitisation to function as an effective funding mechanism for some non-banks (such as insurance companies) as well as banks;
4. to protect investors and to manage systemic risk.

In 2015, the commission made a "Call for Evidence" on the regulation (European Commission 2015). Several responses by interested parties were put forward and grouped in different categories. The main thematic issues consulted included the following:

- affecting the ability of the economy to finance itself and to grow;
- unnecessary regulatory burdens;
- interactions, inconsistencies and gaps;
- rules giving rise to possible other unintended consequences.

(European Commission 2015).

The recent reforms were welcomed by the stakeholders and acknowledged the benefits of the new rules introduced. However, an expression was made concerning the overlapping and inconsistencies between different legislation.

The European Council proposed to the Commission to establish an STS securitisation framework. This STS criteria identifies sound instruments based on a set of established criteria. On the other hand, investors should conduct comprehensive due diligence, to match their investment possibilities with the respective risk appetite. Furthermore, the European Parliament acknowledged that the STS securitisation development can be exploited in a better way. Apart from developing a regulatory framework for high quality securitisation it is advisable that effective methods of risk management should be in place. The European Banking Authority (EBA) highlighted that structured investment products suffered from poor performance and complexity due to the lack of transparency standards prior to the financial crisis European Banking Authority (2014a, 2014c). The one size fits all regulatory approach to securitisation was proved to be no longer appropriate because of the transaction's lenient treatment. Notably, the legislator divided the securitisation regulation into three different chapters, with a different thematic issue addressed separately.

*5.1. Legal Basis for the Securitisation Framework*

A standard pattern is identified, in the regulations and directives enacted at EU level. The EU legislator undertakes the same legal foundation, for the securitisation framework, as it was previously adopted in other regulations and directives. Article 114 of the Treaty on the Functioning of the European Union (TFEU) acts as the legal foundation for the provisions, which is included in the law, regulation and administrative action. The main aim is to improve the functioning of the internal market in this scenario by introducing the STS criteria, and the harmonisation of the existing provisions in EU law. EU legislation is based on two fundamental principles of subsidiarity and proportionality. Indeed,

---

[3] In 2013, the EBA published the Regulatory Technical Standards (RTS) on securitisation retention rules and the respective requirement. Whilst the Implementing Technical Standards (ITS) relates to the convergence of supervisory practices for implementing the additional risk weights in any non-compliance retention rules.

these principles are the foundation of any EU treaty which ensures that powers are exercised as close to the citizen as possible (Panizza 2018). The regulation is based on Article 114 of the TFEU, a legal basis that seeks to establish and maintain the functioning of the internal market. In fact, Article 114 served as a legal foundation for the implementation of the following regulations: Capital Requirements Regulation 575/2013, Credit Rating Agencies Regulation 1060/2009 and Regulation 648/2012 concerning OTC (over the counter) derivatives, central counterparts and trade repositories.

The subsidiarity principle aims to restart a sustainable market by improving the financing of the EU economy, achieved by maintaining financial stability and investor protection. On the other hand, the proportionality principle of the proposed framework focuses on the STS securitisation criteria. This principle includes the responsibility of compliance and notification to European Securities Markets Authority (ESMA). Investors should conduct their necessary due diligence before investing in STS securitisation; however, the supervisory coordination at European level is crucial to ensure that the former is being reinforced. On a general note, the drafting of the securitisation framework is in line with the pre-established definitions and provisions of the EU.

### 5.2. EC Regulation: Establishing Common rules on Securitisation and Creating a Framework for STS Securitisation

### 5.2.1. Regulation Applicability

The regulation applies to selective financial counterparties, which includes: institutional investors exposed to a securitisation transaction, originators, original lenders, sponsors and SPVs (Article 1 of the regulation (EU) 2017/2402 of the European Parliament and of the Council of 12 December 2017 laying down a general framework for Securitisation and creating a specific framework for simple, transparent and standardised securitisation, and amending Directives 2009/65/EC, 2009/138/EC and 2011/61/EU and Regulations (EC) No 1060/2009 and (EU) No 648/2012). Indeed, Article 2, Paragraph 12 of the regulation provides a detailed definition of whom is an institutional investor. Indeed, this includes different types of investors such as insurance undertakings, reinsurance undertakings; institutions for occupational retirement provisions; alternative investment fund manager; UCITS fund managed including internally managed and credit institution. Once the regulation is finalised and implemented, it would overrule the jurisdictional boundaries in the EU member states.

### 5.2.2. Distinguishable Definitions of Securitisation

Different academic authors, as well as regulators, provide different definitions of what securitisation stands for. Article 2(1) of the securitisation regulation defines securitisation as follows:

"securitisation" means a transaction or scheme, whereby the credit risk associated with an exposure or pool of exposures is tranched, having both of the following characteristics:

- payments in the transaction or scheme are dependent upon the performance of the exposures or pool of exposures;
- the subordination of tranches determines the distribution of losses during the ongoing life of the transaction or scheme.

<div align="right">(Article 2 of the Regulation (EU) 2017/2402)</div>

On the other hand, the Article 1 of the Regulation (EC) No 24/2009, which regulates financial vehicle corporation engaging in a securitisation transaction defines it as follows:

- "securitisation" means a transaction or scheme whereby an asset or pool of assets is transferred to an entity that is separate from the originator and is created for or serves the purpose of the securitisation and/or the credit risk of an asset or pool of assets, or part thereof, is transferred to the investors in the securities, securitisation fund units, other debt instruments and/or financial derivatives issued by an entity that is separate from the originator and is created for or serves the purpose of the securitisation, and:

(a)     in case of the transfer of credit risk, the transfer is achieved by the economic transfer of the assets being securitised to an entity separate from the originator created for or serving the purpose of the securitisation. This is accomplished by the transfer of ownership of the securitised assets from the originator or through sub-participation, or, the use of credit derivatives, guarantees or any similar mechanism;

(b)     where such securities, Securitisation fund units, debt instruments and/or financial derivatives are issued, they do not represent the originator's payment obligations.

(Article 2 of Regulation EC No 24/2009)

The Regulation (EC) No 24/2009 definition acknowledges the true sale transfer of the underlying assets. While the definition concerning the other regulation deals directly with the transfer of credit risk of the underlying pool of assets. Indeed, the regulator focuses only on the off-balance sheet securitisation. The on-balance-sheet securitisation falls out of the scope of both the European Central Bank (ECB) and the EU securitisation framework. Of particular importance is the exposure, which creates a direct payment obligation for a transaction or scheme used to finance or operate physical assets and should not be considered as a securitisation exposure. The definition in the securitisation regulation limits the possibility of the on-balance sheet securitisation, which reaffirms the objectives of the regulation in the first place.

### 5.2.3. Mandatory Due Diligence

Given that the nature of structured financial products tends to be quite complicated, the regulation highlights that the related checks and balances should be conducted before any investments are carried out. The regulation acknowledges that if the originator is not a credit institution or investment firm, they must grant credit depending on sound and well-defined criteria. On the other hand, investors must check that the counterparties retain a material economic interest of not less than five percent (5%) and the necessary information should be accessible by the investor on a frequent basis.

Adequate due diligence by the institutional investors must ensure that the necessary risks are being assessed. Exposure to the underlying pool of assets, might include not only credit risks but also a comprehensive list of risks, which might affect the securitisation as well. An assessment of the structural features of the securitisation, which impact the performance of the respective securitisation position, should be included as part of the due diligence. Investors must undertake appropriate due diligence in respect to STS securitisation by consulting the information disclosed by the respective counterparties and ensuring reliance on the STS criteria.

Ideally, institutional investors must establish written procedures commensurate with their risk profile, and which are suitable for their trading. Furthermore, stress tests performance on cash flows and collateral values are vital to determine the complexity of risk of the structured products involved. Internal reporting by senior level management is essential so investors are aware of any material risk arising from the positions to ensure adequate control. These reports are useful when the regulators question the respective securitisation transaction. Additionally, the outlined due diligence ensures that the institutional investor would be able to demonstrate their understanding of their position and the individual underlying exposure.

### 5.2.4. Risk Retention Rules

The maintenance of risk exposure is ideal to ensure the alignment of interest in securitisation, due to the specific banking problems, which occurred not so long ago. Sector-specific regulations included the risk retention requirements indirectly, and the obligation was placed on the investor to check the necessary risk retention. Investors had no direct access to the information to classify the risk retention and perform such checks. The securitisation regulation imposes a direct risk

retention requirement together with a reporting obligation. Conversely, when the counterparties are not established in the EU, the indirect approach would apply.

As already noted, Article 4, paragraph 1 of the regulation illustrates that counterparties should retain on an ongoing basis a material net economic interest of not less than five percent (5%) on an ongoing basis. The net economic interest cannot be split between the retainers, neither hedged, nor subject to any credit risk mitigation. In any case of disagreement, the onus is on the originator to maintain the required amount. Furthermore, only one application of risk retention is allowed for any given securitisation. The notional value shall be determined for the off-balance sheet items. Bavoso (2015a, 2015b), highlights that the risk retention of five percent (5%) is quite controversial. Similar rules used to exist in the past and originators often used to bear the first loss in the lower Securitisation tranches anyway.

The regulation acknowledges a selective list of how risk retention can be classified depending on the value of each tranche sold or transferred to an investor. For a revolving securitisation, the retention requirement should not be less than five percent of the nominal value of the securitised exposure. Other tranches having similar risk profile are factored in, when the first loss tranche does not maintain the established criteria. Furthermore, the regulation acknowledges the retention of the first loss exposure should not be less than five percent of every securitised exposure in the securitisation. A randomly selected exposure should add up to at least five percent of the nominal securitised value, given that the potential securitised exposure must not be less than one hundred at origination.

However, a different approach is illustrated in Article 4, paragraph 4 and paragraph 5, respectively, whereby the five percent (5%) net economic interest would not apply. In fact, the risk retention would not apply for any securitisation exposure backed by a central and regional government or central bank; institutions to which a 50% risk weight or less is assigned (Chapter 2 of the Regulation (EU) No 575/2013 on prudential requirements for credit institutions and investment firms and amending Regulation (EU) No 648/2012), and multilateral development banks). Moreover, the regulation caters for a situation where a credit institution or financial institution is acting as an originator or a sponsor, which is established under the Parent-Subsidiary Directive. Given the relationship, both the parent entity and the subsidiary entity are treated on a consolidated basis for supervision purposes (Article 4, Paragraph 3 of the Regulation (EU) 2017/2402 which complements Article 405 of the CRR, Regulation (EU) No 575/2013). The EBA Report (European Banking Authority 2014b) highlights that the consolidated basis issues are quite challenging for the sponsor (Article 4(2) of the Capital Requirement Regulation). Additionally, the interest alignment is crucial to retain interest on a consolidation basis and not divesting from the group during the transaction maturity.

### 5.2.5. Transparency Rules for Counterparties

The transparency requirement complements the risk retention requirement article. Information should be available without delay by the counterparties both to the securitisation holders and to the respective NCAs free of charge and on time. A quarterly obligation relates to information on the underlying securitisation exposure which should be produced. In the case of an Asset Backed Commercial Paper (ABCP), the data should be provided on a monthly basis.

Another disclosure obligation arises concerning the priority of payments, which should include a set of documents, should be published. In any case, if a prospectus has not been drawn up, an overview of the main features of securitisation must be made available. These quarterly reports must illustrate all material data on credit quality and performance. Furthermore, other critical data to be disclosed are of cashflow nature, which are generated by the underlying exposures. Any liabilities and breaches, which would affect the priority payments, or any replacement of the counterparties, must be published. Collectively, the counterparties must decide which entity is to be in charge of fulfilling the information requirements online through a website. Information should include a well-functioning data quality control system. Appropriate systems must be in place to ensure that the website fulfils its function

reliably and securely. All information should be available for five years, after the securitisation maturity date. A provision for the introduction of draft regulatory and technical standards is also included.

*5.3. Ongoing Transparency Obligations*

5.3.1. The Simple, Transparent and Standard Securitisation (STS) Principle

Bavoso (2013, 2015b, 2015c) highlights that the task of defining securitisation for the STS criteria has been challenging and it still presents some unaddressed questions. The proposed features capture and mitigate the significant risk drivers, as highlighted by the recent financial crisis, which does not directly relate to the riskiness of the underlying exposures. Indeed, the legislator distinguishes between the STS securitisation in general vis-à-vis the STS for the ACBP securitisation. Article 6 ensures that the counterparties shall use the designation of STS and must be in compliance with specific criteria established in the subsequent articles. A notification to the European Securities and Markets Authority (ESMA) shall be made concerning Article 14 sub-paragraph one. An issuer of securitisation shall submit the necessary instrument compliance and assume the responsibilities for any misreporting. Nevertheless, the disclosure requirements should provide the relevant information needed to carry out the due diligence and assess the risks involved. Data required should be available on a website ideally in a standard format and easily accessible (Article 7 of the Regulation (EU) 2017/2402-Securitisation Regulation).

5.3.2. Simplicity Requirements

As a general rule, the STS criteria ensure the safeguarding of the financial world, by introducing requirements and obligations. Article 8 acknowledges that the underlying exposure must be acquired by an SPV, exclusively by a sale or an assignment against the seller or third-party insolvency. In any case of the seller's bankruptcy, there should be no clawback provisions. Furthermore, to enhance the simplicity of the transaction, the seller must provide representations and warranties on the underlying exposure which would not adversely affect the sale or transfer.

Homogeneity issue, related to the pool of underlying exposure, shall be contractually binding and with obligations towards full recourse to debtors. This recourse must illustrate, the periodic payment streams relating to the principal amount and any interest payments due. The legislator highlights that the underlying exposure must not be active portfolio management associated with the underlying pool of assets, based on a discretionary basis. Portfolio management results in an additional layer of complexity leading to an increase in agency risk, affecting the performance of the underlying assets and the transaction. Given the fact, that active management portfolio might involve "cherry picking" of securities, the regulation stresses the importance of satisfying the eligibility criteria of the underlying exposure. The replenishment practice criteria consist in acquiring new eligible assets once cash is received from the securitised assets. Another desirable criterion is the practice of substitution of non-compliant exposures in the transaction which include any breach of representation and warranties which should not be considered as active portfolio management.

The re-securitisation of the underlying securitisation and synthetic securitisation are outside the scope of the simplicity criteria. Both types of securitisation are quite complicated, but the re-securitisation involves a lower credit quality notes repackaged. Given the change in credit performance, it would affect the underlying credit quality. A prime reason for their exclusion pertains to the credit risk modelling and the high correlations involved. Moreover, transferable securities as defined in Markets in Financial Instruments Directive (MiFID) Directive 2014/65/EU are excluded from the underlying exposures of securitisation. These types of financial instruments are considered complex, risky for due diligence analysis carried by the investor.

Insolvency problems, which originated from the originator's business was a standard issue in the last financial crisis. The securitisation regulation seeks to address this problem, especially with the introduction of the STS criteria. Assets transferred through the true sale process should

achieve adequate ring-fencing and segregation. Furthermore, the underwriting standards for the underlying exposures should be similar to the exposures which were not securitised. The importance of disclosing any material changes in the underwriting standard ensures that originators are compliant with transparency criteria. Stringent requirements arise when the underlying exposures are related to residential loans which need to be verified by the lender. On the other hand, the borrower should adhere to the requirements set out in the Directive 2014/17/EU (directive on credit agreements for consumers relating to residential immovable property and amending Directives 2008/48/EC and 2013/36/EU and Regulation (EU) No 1093/2010) in assessing their creditworthiness. An assessment of the borrower's creditworthiness must be undertaken and documented. Once the agreement is in place, the creditor cannot subsequently cancel or alter it to the detriment of the borrower.

When the true sale occurs, the underlying exposures must not be in default. The legislator caters for two scenarios whereby default situations are excluded either under Article 178(1) of Regulation (EU) 575/2013 (regulation on prudential requirements for credit institutions and investment firms and amending Regulation (EU) No 648/2012) or else exposures to a credit-impaired debtor to the originator's best of knowledge. Default defined under the aforementioned regulation is considered when a collateral obligor is unlikely to pay the obligations or any of its subsidiaries in full or recourse. Furthermore, if more than ninety days pass on any material credit obligation to the institution, a default shall be considered. In any case, the competent authority may replace it with one hundred eighty days for exposures related to residential underlying assets. The upcoming securitisation regulation highlights that the above-mentioned exposures should be excluded from the default definition:

- whereby a debtor declared insolvency with the possibility of rescheduling;
- possess an adverse credit history on the official registry of people;
- the debtor has a credit assessment or credit score that the risk of default would be higher than for the average debtor for the type of loan in the respective jurisdiction;
- It is advisable that the debtors must conclude a payment at the time of transfer of exposure from the originator to the SPV.

### 5.3.3. Transparency Requirements

Disclosure requirements have always been a fundamental core element in every European legislation. The main aim is to protect investors and put obligations on the other counterparties, to provide the correct necessary information on time. Documentation and procedures about the underlying transaction should be made available for investors to have a comprehensive knowledge. A prime obligation is the provision of statutory data, in a timely manner, whereby originators and sponsorships need to adhere. Data should be available both in a static and dynamic historical default and loss performance. Nevertheless, the data shall cover seven years for non-retail exposures and five years for retail exposures. External verification by an independent party is required to sample the underlying exposure with a confidence level of ninety five percent for the respective underlying exposure and this shall be published. On the other hand, investors would be able to carry out risk analysis and due diligence which ultimately build investor confidence in the market. The EBA report highlights that the disclosure of loan by loan date on the underlying assets, on a regular basis ensures that the liquidity on the secondary market (European Banking Authority 2014a).

The cashflow modelling in a securitisation deal consists of two parts: the modelling of the cash collections from the asset pool and the distribution of the collection to the noteholders and other transaction parties (Campolongo et al. 2013). The asset side models the cash collection from the underlying asset pool depending on their behaviour. On the liability side, the waterfall model, includes the distribution of cash collection to the noteholder. Another obligation arises to provide a liability cash flow model to investors before the pricing and on-going basis. The liability cash flow model should be available to investors, whereby they can model payment obligations and the securitisation price, respectively.

Article 10 refers to the transparency requirements under Article 5, specifically concerning subparagraph 1(a) to (e). Counterparties are to be jointly responsible for any failure of publishing the information before any pricing of the securitisation. Generally, notifying the investors would include corporate news illustrating the intention of the counterparties in the securitisation transaction. The prospectus must be drafted according to the Prospectus Directive 2003/71/EC and published at least two weeks in advance, prior to the closing of the transaction. Hence investors would have access to all information to make the necessary informed investment decision.

### 5.3.4. Standardisation Requirements

A securitisation transaction is very complicated. However, the securitisation regulation seeks to ensure that the standardisation principle is given utmost importance. Counterparties must adhere to the risk retention as outlined in Article 4. Standardisation of risk must be minimised and mitigated, primarily for interest rate risk and currency risk. In fact, these risks are to be disclosed according to the transparency requirements under Article 5. Derivatives should be solely included only to hedge the risks as mentioned above. While interest payments should be based on market interest rates and should not involve any complex formula or derivatives.

Certain securitisation transactions could trigger an early amortisation event, which is used to accelerate the payment of the bond principal. If one of these events trigger the initial amortisation event, it must be disclosed in the transaction documents. The legislator outlines the obligations and responsibilities, which are placed on the servicer and the respective management team leading the securitisation. Another specification involves how processes and responsibilities are handled concerning to any default and the insolvency of a servicer. Nevertheless, the replacement of derivative counterparties or liquidity providers, due to any negligence or any other specific event should also be documented. These standards would ensure that the investors have certainty over any counterparties' replacement involved in the securitisation, and which impact the credit risk. In this respect, policies and adequate risk management are highly encouraged and must be well documented.

The regulation suggests that a trustee must be appointed which ensures that fiduciary obligations are in the best interest of the investor. The legal documents must provide information on how any conflicts may occur and how to resolve any disputes between the noteholders.

### 5.4. The Asset Backed Commercial Paper Securitisation: Similarities and Distinctions to the Term Securitisation

Securitisation transaction maturities can range from a short-term exposure to long-term exposure. ABCP is considered to be a short-term investment vehicle with a maturity of 90 or 180 days and generally issued by a financial institution (Choudhry and Fabrozzi 2008). A pool of homogenous assets must back the requirements for an issue of ABCP programme. Contrasting the other type of securitisation, the ABCP must have an average life of not more than two years and with a maturity of no longer than three years. Residential, commercial mortgages and the transferable securities are excluded from the ABCP. For calculation of interest payments, complex formulae or derivatives are excluded and only the market interest rate is acknowledged as the referenced interest payment.

The underlying exposures shall originate from the sellers' business only. Similar to the simplicity criteria, any material changes in the underwriting standards should be disclosed to the investors. Both transaction document requirements for ABCP and term securitisation are similar however with some exceptions. Indeed, the documents should include provisions for any failure to generate new underlying exposure in to meet the credit quality. The sponsors for securitisation are only limited to credit institutions, in accordance with Directive 2013/36 EU, providing liquidity to the securitisation structure as well as covering securitised exposures. Sponsors must carry out the necessary due diligence and assess whether the seller meets the sound underwriting standards complying with requirements highlighted in Regulation (EU) No 575/2013. Article 259, subparagraph 3 (i) to (m) illustrates how the underwriting is maintained by the asset purchase together with the liquidity facilities and credit enhancements. The programme must include a risk mitigation strategy with respect

to the seller's performance. One notices that the ABCP program resembles the STS criteria, with some distinctive issues. The simplicity criteria are addressed in Article 13, whereby re-securitisation and credit enhancement have been excluded from the tranching. All counterparties are obliged to abide with the risk retention pertaining to Article 4. Given the complexity of a call option, these types of derivatives are excluded from the structuring of the securities in the ABCP programme.

## 5.5. The STS Notification Procedure

ESMA is in charge of the STS notification provided by the counterparties, in compliance with the established criteria listed in Articles 7 to Article 10 concerning term securitisation, or Articles 11 to 13 for the ACBP programme. A list of all securitisations shall be maintained and published on the ESMA website on a frequent basis. Ideally, there should be a point of contact between the counterparties with the competent authorities to communicate with the competent authority. In any case that the counterparties no longer maintain the STS criteria; it is advisable that they notify ESMA immediately to update any records.

If the originator is not a credit institution as defined under Regulation 575/2013, it must confirm that the credit granting, is based on sound and well-defined criteria. A declaration should include established criteria for approving, amending, renewing and the financing of credit. In any case of administrative sanctions imposed by the NCA's, EMSA should be notified immediately for publication of the respective sanction. With respect to the STS notification procedure, ESMA will be in charge of drafting the necessary regulatory and technical standards, a year post-implementation date and shall be forwarded to the commission.

## 5.6. The Role of Competent Authorities in a Securitisation Structure

The role of the national regulator deals with thematic issues addressed throughout the securitisation regulation. Compliance with the due diligence obligations must be according to the relevant legal acts which regulate the respective entities. In case that the entities' activities (Article 3 of the Regulation (EU) 2017/2402) fall outside the scope of any directive or regulation, the competent authority should inform the commission and the European Supervisory Authority (ESA), respectively. On the other hand, sponsors must comply with the obligations set out in Articles 4 to 14 of the regulation (Article 15 (2) of the Regulation (EU) 2017/2402). Indeed, ESMA will be in charge of maintaining an updated list of the NCA's.

NCA's in the respective member states are empowered to review and assess the securitisation structure implemented by the counterparties, mostly concerning risk issues which affect the counterparties. Another responsibility given to the NCA's is to ensure that the risks are being evaluated and addressed properly through the use of effective risk management. The regulation is concerned about the importance of managing reputation risk and intends to avoid events, which happened during the recent financial crisis. A classical case which made shocking worldwide headlines, in 2007, was the Northern Rock bank run. It relied heavily on the wholesale market rather than the retail market with an aggressive business model. This model proved profitable when the market prospects were good. However, the reverse happened when the excessive subprime lending occurred, and the bank was unable to meet the obligations. Given the fact, that Bank of England had to intervene as the "lender of last resort", the bank's reputation was severely tarnished. Indeed, this led to the first bank run in the UK for more than one hundred and fifty years. Appropriate policies must be implemented by the counterparties aiming to minimise and manage effectively the risks involved in securitisation.

Administrative, Criminal Sanctions and Remedial Measures

Any breaches, which result from a failure of obligations concerning the regulation, will be administered by the respective NCA's. Member State shall establish rules together with appropriate administrative sanctions about:

the failure to meet requirements in respect of risk retention (Article 4 of the Regulation (EU) 2017/2402);

the failure to meet requirements in respect to transparency (Article 5 of the Regulation (EU) 2017/2402);

the failure to meet requirements of the STS criteria or the ABCP programme (Article 7 of the Regulation (EU) 2017/2402) (Article 11 of the Regulation (EU) 2017/2402);

The NCA's is empowered to exercise their duties with respect to the administrative sanctions and remedy measures following their national legal framework. Collaboration with other authorities and competent juridical authorities is highly encouraged. Before sanctioning the counterparties, the competent authority should evaluate the relevant circumstances impinging on the respective infringement. In fact, these circumstances can range depending on the different factors such as materiality and duration; the natural or legal person involved, and profits or loss involved. Most compelling situations revolve around the actions of the natural and legal persons and their financial strength, in particular, the total turnover. The latter revolves around the total turnover or else the annual income and net assets of the natural or legal person. Cooperation level by the responsible natural or legal person with the competent authority must be ensured to the maximum level, without any discharge of profits or losses avoided. Article 19 of the regulation includes a provision referring to any member states which does not decide to implement any sanctions. Any sanctions given should ensure that the appropriate measures are in place for the competent authorities to exercise their power and the necessary information must be passed on to the ESA to fulfil their cooperation obligation.

Regulation (EU) 2017/2402 provides a list of sanctions for any failure on the above-mentioned requirements. Administrative fines are a universal sanction due to a failure of obligation, which is at least of EUR 5 million or the corresponding value in any other currency. In case of a legal person, it could either result in an administrative penalty of EUR 5 million or up to ten (10%) percent of the total annual turnover of the legal person depending on the last available financial accounts. Administrative fines can be imposed on a parent-subsidiary relationship, taking into consideration their yearly turnover. A maximum administrative fine is set at twice the amount of either EUR 5 million or else the established ten percent of the total annual turnover. On the other hand, the competent authorities can issue a public statement indicating the identity of the natural or legal person and the infringement according to the publication of administrative sanctions. Another sanction involves a temporary ban against the counterparties and their management body which is responsible for the company. A temporary ban would result from any infringement by the counterparties to the fundamentals of this regulation (the STS criteria and the ABCP) with the right to appeal from any administrative sanctions before the tribunal level.

*5.7. Adequate Cooperation between EU Regulatory Bodies and the National Competent Authorities*

The European regulatory system consists of three European Supervisory Authorities, introduced by the Lamfalussy process way back in 2001. These authorities include the European Banking Authority (EBA), European Insurance and Occupation Pensions Authority (EIOPA) and European Securities and Markets Authority (ESMA). An important concept introduced, is the exchange of information with other competent authorities, to collaborate with other competent authorities, particularly in identifying any infringements. In case of any regulatory breach, the competent authority must inform and co-operate with the other respective competent authority to ensure coordination and consistent decisions are taken. Moreover, the ESA must be notified of essential findings in infringement, especially about the STS notification. On the other hand, any criminal sanctions shall remain published for at least five years term. The maintenance of central database for any administrative sanctions and remedial measures is an essential method for exchange of information amongst the competent authorities.

*5.8. The Amendment of the EU Regulation for Capital Requirements Regulation No 575/2013*

Another prominent reform following the post-financial crisis included the amendment to the several regulations and directives, at EU level. The CMU action plan is based on two legislative measures: the securitisation regulation and the amendments to the regulation of capital requirements. In the following sections, the author highlights the salient capital treatments concerning securitisation. The objective is to introduce a more risk-sensitive regulatory treatment for the STS securitisation. It complements the previous aim of developing a common substantive framework for securitisation.

On an international level, one should acknowledge Basel's work on the adequate allocation of capital requirements. On the other hand, on a European level, the EBA is in charge to report on the qualifying securitisation, with a recommendation of lower capital charges for STS securitisation. Furthermore, the commission took the initiative to amend the Capital Requirements Regulation (CRR) No 575/2013 European Banking Authority (2014b, 2014c), by adjusting the risk retention profiles equipped with proper features for the STS securitisation. This regulation overrules the jurisdictional boundaries between EU Member States and non-EU Member States.

Chapter five of the CRR is dedicated to securitisation and illustrates that this regulation is in line with previously mentioned international regulation. Adequate capital for securitisation is dependent on two crucial factors: the qualitative conditions and the quantitative factors. Ideally, these factors ensure in maintaining the minimum levels of credit quality while ensuring the STS principle does not finance risky underlying exposures. Prudential capital requirements were introduced in several articles, notably Article 260, Article 262 and Article 264. These approaches ensure a lower capital charge for the senior positions in securitisation. While the non-senior position will still be charged at a higher rate of fifteen percent (15%). The STS securitisation criteria seek to reduce the model and agency risks. However, it must fulfil some essential requirements such as the credit-granting standards, granularity, and maximum risk weights.

Quantitative and Qualitative Factors for Determining the Adequate Capital Requirements

Notably, the maximum risk weight plays an important role in senior securitisation positions. A look through approach is adopted for any securitisation position, which receives a maximum risk weight equal to the average risk weight applicable to the underlying exposures (Delivorias 2018). This approach also applies for securitisation positions that are rated or unrated. Even if a different approach is used for determining the underlying exposure, the credit institution must determine the credit risk exposure.

A maximum risk weighted exposure cap is planned for institutions to calculate the K IRB. Whilst retaining this treatment of applying the maximum capital requirement, which is equal to the capital requirement of the non-securitised exposure (Article 260 of Regulation No.575/2013 of the European Parliament and of the Council of 26 June 2013 on prudential requirements for credit institutions and investment firms and amending Regulation (EU) No 648/2012.). With respect to the SEC–ERBA and SEC–SA, the former treatment is engaged, given that the securitisation is portrayed as a credit risk mitigation. The CRR regulation includes treatment for specific exposures such as for example the SEC–SA methodology must be used for a revolving securitisation. Special treatments for certain exposures with a special reference to the second loss positions in ABCP securitisation and own funds securitisation for revolving exposures are to be eliminated.

With respect to the qualitative factors, the underlying exposure must have at a least a minimum level of credit quality. This is achieved by the underlying pool being sufficiently granular. Any exposure related to a group of connected clients is considered as a single source of granular. From a prudential perspective, all underlying exposures must be established from an EEA jurisdiction and meeting specific conditions under the SEC–SA and the eligibility for credit risk mitigation.

## 6. Methodology

### 6.1. Case Study Approach

To reach our goals and answer the above noted research questions we analysed academic documents including academic papers, academically researched printed chapters, academic journals, articles and monographs, Financial Services Rules, industry guidelines, recommendations, EU directives and regulations. The authors make use of the case study methodology, as suggested by Yin (2003, 2014) and Yazan (2015), on Malta. That is, the authors followed a set of pre-specified procedures. These pre-specified procedures revolved around the following three factors:

- the research question;
- the extent of control over the actual behavioural events;
- the degree of focus on contemporary as opposed to historical events.

The advantage is that of studying the case in its natural setting, through observation of the actual practice (Meredith 1998) and (Ebneyamini and Sadeghi Moghadam 2018).

### 6.2. Interview Procedure and Sampling

The Authors conducted structured interviews with industry experts to collect the necessary data for this study Saunders et al. (2009). Zammit and Grima (2018) states that "To provide the most reliable conclusions to the research question, the author required to delve into the original source of the data being researched".

Using a non-probability-purposive sampling we listed prospective participants. This list included personnel from the Malta Financial Services Authority (MFSA), Finance Malta, the Malta Association of Risk management (MARM), The Malta Association of Compliance Officers (MACO), and other financial industry practitioners who were actively involved in the securitisation regulation process.

Once saturation was reached at 32 interviewees and no more value was received from an extra interview, no more interviews were carried out. Furthermore, the participants provided extensive information on the subject being studied and the authors believe that their knowledge and experience satisfied the research questions outlined previously (Morse 2015).

Although open-ended, all interviews where structured along the following themes:

The characteristics of the Maltese securitisation market;

The challenges and impact of the securitisation regulations;

The STS procedure;

The Capital Requirement Framework;

Operational aspects;

The Capital Market Union.

### 6.3. Analysis

All data from the case study and the interviews were subjected to a thematic analysis as suggested by Braun and Clarke (2006) to determine the main common themes on the reasons why this securitisation regulation is needed locally, the type of requirements and costs this regulation will bring about and the implications of the securitisation regulation for Malta. These were highlighted and discussed under the next sections to maintain the flow.

We then subjected our findings to a comparative analysis; as suggested by Yazan (2015); with other European jurisdictions, specifically Luxembourg and Ireland. This was a suggestion made by all participants interviewed. They suggested that although these jurisdictions might be geographically different from Malta, their economy is considered to be small vis-à-vis other European jurisdictions. All three EU states have developed their securitisation legislation to suit the respective market needs. Both Ireland and Luxembourg attract high volume transactions in auto loans and credit card securitisation.

*6.4. Limitation*

While conducting the necessary research for this paper, we encountered certain limitations. One limitation was the limited amount of expert knowledge and experience with this regulation given that the regulation had only been in force for a few months when the study was being carried out. However, we tried as much as possible through our contacts, to mitigate this limitation by carrying interviews with participants who had been working with this regulation as consultants, regulators, associations and lobby groups before and after the development, consultation and implementation stages.

## 7. Malta as a Securitisation Jurisdiction

*7.1. The Legal Background*

Prior to the introduction of the securitisation regulation, the domestic legislation was based on three legislative acts, which were broad in scope:

The Securitisation Act Chapter 484 of the Laws of Malta, which regulates the SPV's operation and their respective securitisation transaction.

The Subsidiary Legislation 386.16, the Securitisation Cell Companies Regulation outlining the creation, operation and the notification of cell structures provides an exhaustive list of assets securitised, from a domestic market (Malta) perspective.

The Subsidiary Legislation 403.19, the Reinsurance Special Purpose Vehicle Regulation, whereby securitisation vehicles are established as reinsurance special purpose vehicles to assume insurance risk and issuing insurance-linked securities subject to approval from the MFSA.

*7.2. Chapter 484—The Securitisation Act*

The Securitisation Act enacted in 2006 provides an attractive legal, regulatory and tax framework for the securitisation vehicles incorporated in Malta. A securitisation vehicle can be established as a company, partnership, trust or other legal structure, as permitted by the MFSA. Securitisation vehicles are not treated as a collective investment scheme, under the act.

An important provision outlined in the act is that a securitisation vehicle is treated separately from the insolvency proceedings once the assets have been transferred by the originator. A securitisation vehicle achieves bankruptcy remoteness by orphaning the securitisation vehicle. This can be done through either a Maltese charitable trust or a Maltese purpose foundation. Hence, the originator can retain an equity stake in the securitisation vehicle.

An assignment of receivables to a Maltese securitisation is considered to be final, absolute and binding on the following parties: the originator, the securitisation vehicle and all third parties. It is important that the governing law of the receivables and the instrument of assignment should be referred. Any data or information transferred in a securitisation should be completed in line with the Data Protection Act (Chapter 440 of the Laws of Malta).

Investors are granted an assignment by way of security of the issuer's rights and a pledge of cash accounts of the issuer. Furthermore, the Securitisation Act grants statutory privilege concerning the securitisation assets and which extends to the proceeds derived by the securitised assets. In fact, this privilege ranks ahead of all other claims at law and expect for securitisation creditors, who enjoy a prior ranking. The act acknowledges that the right of various securitisation creditors to contractually regulate their ranking between themselves.

Securitisation vehicles are able to eliminate any tax leakage any achieve tax neutrality with a combination of:

- Deductibility of expenses provisions under the Income Tax Act.
- Deductions specifically under the Securitisation Tax Rules.

Chargeable income deductions may result in no income tax payable in Malta. However, if the securitisation vehicle has any remaining income after deducting all allowable expenses, there is a possibility to claim a further deduction of an amount equal to the remain income. With respect to taxation of investors, no Maltese tax is withheld/payable on payments of interest by a securitisation vehicle to a holder of the vehicle's securities provided that the investor is non-resident, does not have a permanent establishment in Malta or is not owned and controlled, or acts on behalf of an individual who is ordinarily resident and domiciled in Malta.

The act distinguishes between both public and private securitisation vehicles, whereby the former must be licensed by the MFSA to issue financial instruments to the public on a continuous basis. Private securitisation vehicles are obliged to notify the MFSA prior to commencing business and must submit a notification form outlining the corporate information and the details relating to the securitisation transaction/s. Moreover, the private securitisation vehicle is considered to be a Financial Vehicle Corporation for the purpose of Regulation (EU) 1075/2013. Every quarter the private securitisation vehicle must submit a quarterly statistical report on its assets and liabilities to the Central Bank of Malta.

### 7.3. An Analysis of the Maltese Securitisation Market

Although the securitisation market is still in its infancy stage when compared to larger jurisdictions, whereby the volume and quantity are much higher, Malta has become one of the fastest growing jurisdictions in the Europe, together with other established jurisdictions such as: Switzerland, Luxembourg, Germany and France. This success, as corroborated during the interviews with participants, is mostly due to the following factors:

- No restriction with respect to the underlying assets.
- Tax neutrality achieved through deductions from the Income Tax Act and the Securitisation Act, respectively.
- No value added tax is applied to transactions.
- Issuance of securities issued on both the regulated and non-regulated market.
- No licensing requirement or authorisation.
- Non-EU licensed fund managers may use securities issued by the securitisation vehicles backed by units in non-EU funds, as a route to accessing finance within the EU.
- Listing of asset-backed securities on the local stock exchange: the European Wholesale Securities Market (EWSM) and the Institutional Financial Securities Market (IFSM).
- Bankruptcy remoteness for the SPVs.
- Robust legal framework for securitisation whereby the securitisation cells are distinct and separate from each other.
- The introduction of the Protected Cell Company structures used in insurance securitisation transaction.
- Securitisation structures are highly competitive for securitisation arrangements below EUR 100 million thresholds.
- Transfers to SPVs of securitisation assets are final which cannot be challenged or characterised.
- Simplification pertaining to the legal formalities for transfer of securities.
- Investors and creditors are granted preferred claim by law.

The most popular assets for securitisation transactions through Maltese vehicles are not concentrated in one industry sector. However, the maritime, aircraft and property sector remain popular.

### 7.4. An Analysis of the Maltese Regulatory Approach

The MFSA, as the competent authority is in charge of the notification procedure of the securitisation vehicles. Since March 2018, the Conduct Supervisory Unit is in charge of the Securitisation Regulation

instead of the Securities and Markets Unit. In 2015, this former department was set up for securing the appropriate consumer protection and addressing any potential risks in financial services. The Conduct Supervisory Unit supervises the corporate services providers and trustees' providers. Indeed, the authors opine that this change in the department is in line with the main securitisation regulatory objectives outlined. Nevertheless, the tasks carried out by the Conduct Supervisory Unit are somehow related to securitisation, especially the trustee's services, which are also part of the securitisation transaction. The Conduct Supervisory Unit trained its employees on the regulation. Furthermore, we believe that the EU Securitisation Regulation will raise the standards in the industry, which will be a gamer changer in the nearby future.

The grandfathering provision is not considered as a hurdle for the MFSA to take control. Hence, it will not affect so much the regulator's work in this aspect because few securitisation issues only date back to 2011. In this respect, one needs to keep in mind, that the legal framework in Malta was introduced in 2006 and was not so popular back then, with a popularity peak only in 2015 and 2016, respectively. This peak indicates that certain category license holders have incorporated both SPVs and Securitisation Cell Companies (SCC)s to issue debt securities and offer a different type of service to their professional investors. The regulator needs to ensure that a balance is struck for the previous transactions, which should be in good faith and without disrupting the market in general.

Another significant issue which the MFSA needs to factor in is the responsibility for authorising third-party verifiers. Due diligence assessments will be carried out by the NCA, whereby these third parties may also be subject to the local legislation. The regulator must carry out the necessary checks and balances that the eligible counterparties are adhering to the STS criteria, and must be established in the EU. Given this, the MFSA might issue a set of binding rules whereby the verifies need to comply.

We however, believe, that even after discussion with sponsors of companies choosing Malta as the European Jurisdiction to licence their company, that these are not the only reason. Other factors such as a good technological infrastructure which permits good communication, climate, a strong workforce with a good level education and experience in the financial services sector, a strong command in languages especially the English Language, the country's position in the centre of the Mediterranean and although still of a high quality when compared to larger jurisdictions, lower legal and audit fees. Moreover, this popularity of Malta as a chosen jurisdiction is also a result of a flexible but strong regulatory structure and regulator, open to promote and work closely with the industry.

However, lately this popularity has been somehow tarnished by some scandals in the Banking, Gaming and political scenarios. Both the Malta Financial Services Authority and the Financial Intelligence Analysis Unit have increased their regulatory scrutiny on all the financial sector. This coupled with the lack of depositories on the island and the de-risking process by the banking sector has made it harder and more expensive for investors to open up shop in Malta.

Although the securitisation regulation is set to be a game changer for the financial market in the EU, the regulator needs to ensure that US influences in the European securitisation are no longer in existence. Tranching, created significant risks to the financial markets and the securitisation regulation is intended to limit this effect. However, from a Maltese perspective, the transactions carried out are both low in value and amount.

*7.5. A Comparative Analysis of Securitisation in EU Small Jurisdictions*

In 2004, Luxembourg introduced the securitisation Law of 22nd March 2004. A Luxembourgish SPV is an unregulated entity and is not subject to any authorisation nor supervision. However, SPVs which issue securities to the public on a continuous basis must be authorised by the "Commission de Surveillance du Secteur Financier". Luxembourg offers quite a broad range of securitisation product offering which focuses on real estate, private equity, Islamic finance, non-performing loans and peer

to peer lending[4]. Similar to the Maltese securitisation market, the majority of the Luxembourg SPVs are established to service foreign originators within the EU. Luxembourg created a flexible legislation concerning the securitisation and included the fundamental elements: investor protection principles, bankruptcy remoteness, protection against insolvency and recourse, which are similar to the principles established in the Maltese legislation.

Ireland's securitisation legislation is outlined in Section 110 of the Tax Consolidation Act 1997. A distinguishable feature includes that the SPV entity must be a resident in Ireland with qualifying assets of at least EUR 10 million, to qualify for tax efficiency purpose. Tax benefits include a value added tax exemption and interest expense deductions from profits. Another tax exemption concerns the listing of debt securities classified as "quoted Eurobonds". However, a specific criterion needs to be met. Not withholding tax is charged given that the paying agent is outside of Ireland, the securities are held on a clearing system, and the investor is a non-Irish resident. Favourable tax treatment might be possible if the interest paid to a resident in the EU or in the other country where a double taxation treaty exists with Ireland.

Ireland has an established regulated stock exchange for the issuing of debt securities. The Irish jurisdiction offer a much broader securitisation market than Malta which include medium-term notes, US life settlements, loan participation notes and distressed debts. Overall Ireland's securitisation offering focuses more on the taxation side, which is quite different from both Malta's and Luxembourg's holistic offering, which focuses on different aspects in a securitisation transaction.

When Malta is compared to both Luxembourg and Ireland as a securitisation jurisdiction, the key advantages concerning corporate, legal, and accounting services are mostly attractive to the securitisation arrangements below the EUR100 million level. Through the years, Malta made significant improvements in the securitisation market, mostly evident by the introduction of legislation. It has substantial opportunities for further growth in the securitisation market. We believe that Malta is considered as a niche jurisdiction for smaller transaction and there should be increased visibility of Malta as a jurisdiction for securitisation through active participation in bespoke international conferences and tactical public relations events. Another important factor is leveraging the strength of the existing aviation and maritime sectors by providing solutions for these markets. It is important to acknowledge that the securitisation structures established in Malta are not influenced by the US securitisation exposures like other EU jurisdictions.

## 8. Salient Points, Conclusions and Recommendations

Due to its operational nature, the shadow banking phenomena overtook the financial world, following a surge in the housing market in the US. Unfortunately, this spread to the European financial market due to the interconnectedness of financial institutions. Indeed, international regulators' intervention kicked in immediately as the financial crisis broke.

On an international level, the Basel committee outlined the framework for securitisation by introducing the STC criteria for securitisation. US regulators tackled the issue of securitisation in the Dodd–Frank Act. Conversely, the EU had lingering problems related to the banking crisis leading to the sovereign crisis. The plan for the revival of the market was initiated in 2015 for the development of the CMU and introducing the securitisation regulation. The plan took a bit longer than expected to be implemented. The fundamental principles of this regulation are based on the STS criteria, which ensures the counterparties are compliant with a set of obligations. This regulation is specifically targeted to institutional investors and not to retail investors, which stresses the importance of monitoring risks on an ongoing basis.

Information related to the criteria is collected by the NCAs acting as repositories to the European Supervisory Authorities (ESA). Moreover, the NCA's have the administrative powers to sanction and

---

[4]    A debt financing method whereby the credit institution is eliminated from the structure.

fine any securitisation structure, which is not in line with the obligations. High burden penalties range from EUR 5 million to ten percent (10%) penalty of the financial income of the preceding year's latest accounts.

A risk-sensitive approach is adapted to calculate the adequate capital measures for securitisation structure, which is addressed in the CRR No.575/2013. When the securitisation structure classifies for the STS securitisation criteria, this eligibility leads to a lower capital requirement. Indeed, a capital requirement of fifteen percent (15%) should be charged. A retention of five percent related to the risk retention by the originators should be retained in a securitisation structure. This retention ensures that the counterparties would retain their minimum interest in the structure.

This regulatory approach together with a capital market plan is set to bring about the necessary changes, which have long been desired. In fact, European regulators want to ensure that American influences in securitisation are history. It is our opinion that the securitisation legislation is undoubtedly working efficiently, even though this regulation increased the cost of compliance both for the NCA's and the counterparties. This is the road to recovering and building confidence in the securitisation market. Nevertheless, the NCA's need to ensure that the necessary training is provided to the employees to carry out the required regulatory obligations. Moreover, it is importance that the counterparties maintain risk management principles since this would prove beneficial to the securitisation structure.

The European securitisation effectively boosts lending to European businesses and households as well as broadens investment opportunities. It provides a harmonised securitisation framework for all securitisations, replacing the prior piecemeal framework spread out in multiple pieces of legislation. All institutional investors are now subject to the same securitisation rules, creating a level playing field, which is simple, less risky, transparent and standardised.

Furthermore, the introduction of the new debt securities market should encourage the growth in the non-traditional lending market, especially for the small and medium-sized businesses. However, this development should be supported by considering the investor protection principles and focusing on financial education of the risk involved in securitisation.

The EU's securities markets regulator, has launched a consultation on draft Regulatory and Implementing Technical Standards (RTS and ITS) under the Regulation (EU) 2019/834 (EMIR REFIT) covering reporting to Trade Repositories (TRs), procedures to reconcile and validate the data, which are accessed by the relevant authorities and registration of the TRs (European Parliament and of the Council 2019).

Though this, as highlighted by all interviewees, has and will increase transparency, they all envisaged that the regulatory costs are considered to be seen as a burden to adhere with established rules. Moreover, the new regulation will have a limited effect on Malta, given that the regulation focuses on tranching securitisation, which has a higher value, both in terms of volume and amount.

Recently, Malta made significant improvements in the securitisation sector, mostly evidenced by the introduction of the legislation and other factors highlighted in the previous sections. All interviewees emphasised that Malta has substantial opportunities for further growth in the securitisation market and it is encouraged to be exploited well.

We agree with the latter statement and believe that Malta has a lot to offer with its strong financial background, experience and reputation, which although have been tarnished of the late, still offer great potential for the financial services industry. Malta needs however to attract more depositories or lobby with the European Securities and Markets Authority (ESMA) to consider allowing passporting of EU licenced depositories, maybe also considering proportionality (Treaty of Lisbon 2007). Moreover, the MFSA should carry out more consultation with the Maltese industry players and banks to determine the implications of the de-risking process, since it might be the case that this is creating more risk than good and evidently as Grima et al. (2020) suggests is not reaching the required or expected goals.

However, on the flip side we agree that, the regulatory implication on the domestic market will have a limited effect a since the regulation focuses on the tranching securitisation which involves a higher value, both in terms of volume and amount. Moreover, the robust legal framework for

securitisation whereby the securitisation cells are distinct and separate from each other and the introduction of the Protected Cell Company structures used in insurance securitisation transaction can offer great advantages for setting up in Malta.

Given this, we feel that it would be interesting to carry out a follow-up research on this EU regulation post-implementation date since we feel that securitisation will encourage the growth in the non-traditional lending market, especially for the SME business. However, this development should be supported by considering the importance of investor protection principles. We opine that both the competent authorities and the counterparties should cooperate together to ensure that the regulations principles are being safeguarded.

**Author Contributions:** Supervision, S.G.; Writing—original draft, J.M. Writing—review & editing, S.G., S.S., R.R.-A. and M.L.Z. All authors have read and agreed to the published version of the manuscript.

**Funding:** This research received no external funding.

**Conflicts of Interest:** The authors declare no conflict of interest.

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
