# Peer review of "A Study of the Implications of the European Securitisation Regulation 2017/2402 on Malta"

_laws, 2017_

Round 1

Reviewer 1 Report

The purpose of this manuscript is to investigate the international regulation, but i do not see How Do this paper contribute to the existent literature in this form. What is new?   However some questions arising and maybe they require paying attention for the authors, as follows:   First: you mention in abstract: ”Document analysis, as a form of qualitative research, is used in this study.” Where is the methodology? Where are the concludend section: conclusions?

Second: In the introduction I suggest writing a paragraph with the sections of the paper.

3. The limitations of the paper should be also analyzed in ”Conclusions”.

Minor comments

1.Title ”The Implications of the European Securitization Framework to Ensure a Harmonized, Simple and Regulated Securitization Market” appears to be so long. I suggest to restrict it to be more concise. For instance:There is a real need for a new EU regulation?

2. The language and grammar should be reviewed. There are also several incomplete sentences.

Author Response

Reviewer 1

Thank you very much for your time and valuable input, which has helped to make our paper better. We have made the amendments in the paper following your comments and suggestions, as can be noted in bold below your comments.

The purpose of this manuscript is to investigate the international regulation, but i do not see How Do this paper contribute to the existent literature in this form. What is new?   However some questions arising and maybe they require paying attention for the authors, as follows:   First: you mention in abstract: ”Document analysis, as a form of qualitative research, is used in this study.” Where is the methodology? Where are the conclude end section: conclusions?

We narrowed our study to our original aim, which was that of an analysis of the main provisions of the Regulation No. 2017/2402 on Malta as a jurisdiction for securitisation and provide an insight on the prospective market development.

We have included a section on the methodology section (6) and adjusted the last section (8) to include the Salient points, conclusion and recommendations section.

The limitations of the paper should be also analyzed in ”Conclusions”.

We have included the limitations in (8) Salient points, conclusion and recommendations section.

Minor comments

Title ”The Implications of the European Securitization Framework to Ensure a Harmonized, Simple and Regulated Securitization Market” appears to be so long. I suggest to restrict it to be more concise. For instance: There is a real need for a new EU regulation?

The title was changed to – ‘A Study of the Implications of the European Securitisation Regulation 2017/2402 on Malta’. We feel this is more appropriate.

The language and grammar should be reviewed. There are also several incomplete sentences.

The grammar and sentences were reviewed by native English speakers.

Reviewer 2 Report

  • The paper is merely descriptive, without deploying a critical perspective nor addressing any critical aspects of the framework; too long for what is conveys.
  • Furthermore, the declared objective of the paper is only partially fulfilled: 1)“With this article, we aim to highlight through a rigorous analysis of the international, the US and the EU regulatory framework and a literature review the implication of introducing the new EU regulation and provide an insight on the prospective market development”: however a) the US regime is not analyzed in depth (only a short paragraph and only through secondary sources); b) the analysis of EU regime lack references to the last developments (see ESMA 2019 RTS/ITS); 2) “Ultimately, we demonstrate that the new EU regulation has provided for an effective and efficient Securitization market by ensuring that investors are protected by strict compliance to the STS Securitization criteria and proper overall supervision”: I could not find any ‘demonstration’ (i.e. at least reasoning, comparison with US, data, etc.); 3) the reference to the NCA issue mentioned in the abstract is not discussed adequately in the text. I therefore suggest to declare in the abstract the merely descriptive nature (or, better, improve the analysis with critical aspects) and eliminate the US part;
  • Other aspects to be improved:
    • I would add some discussion about the need of securitizations (considering the financial crisis negative effects, why should we encourage securitizations), besides references to the EC documents
    • Sources/references: many pillars in the literature about securitization have not been mentioned/analyzed (see Iacobucci, Kettering, De Vries Robbé, Jobst, etc.) as well as other recent papers/contributions on the topic
  • Minor changes required:
    • In the introduction, report data about volumes before and after the crisis (EU reforms also aim to foster the market again)
    • 2 “It is our opinion that the EU Securitization legislation is now, so far working efficiently with this Securitization regulation, even though this regulation d the cost of compliance both for the NCA's and the counter-parties”: sentence unclear/with errors
    • Paragraph 2: need to mention the originate-to-distribute problem
    • 3, § 2.1: It seems that you are saying that the IRB method was introduced with Basel III
    • 8 first paragraph: put the following sentence in a footnote instead of in a parenthesis, please: “European delegated regulation together with a significant number of regulatory and implementing technical standards (In 2013, the EBA published the Regulatory Technical Standards (RTS) on Securitization retention rules and the respective requirement. Whilst the Implementing Technical Standards (ITS) relates to the convergence of supervisory practices for implementing the additional risk weights in any non-compliance retention rules”.

Author Response

Reviewer 2

Thank you very much for your time and valuable input, which has helped to make our paper better. We have made the amendments in the paper following your comments and suggestions, as can be noted in bold below your comments.

The paper is merely descriptive, without deploying a critical perspective nor addressing any critical aspects of the framework; too long for what is conveys.

We narrowed our study to our original aim, which was that of an analysis of the main provisions of the Regulation No. 2017/2402 on Malta as a jurisdiction for securitisation and provide an insight on the prospective market development.  We have addressed this in a new section (7) Malta as a Securitisation Jurisdiction.

  • Furthermore, the declared objective of the paper is only partially fulfilled:

1)“With this article, we aim to highlight through a rigorous analysis of the international, the US and the EU regulatory framework and a literature review the implication of introducing the new EU regulation and provide an insight on the prospective market development”: however a) the US regime is not analyzed in depth (only a short paragraph and only through secondary sources); b) the analysis of EU regime lack references to the last developments (see ESMA 2019 RTS/ITS);

We narrowed our study to our original aim, which was that of an analysis of the main provisions of the Regulation No. 2017/2402 on Malta as a jurisdiction for securitisation and provides an insight on the prospective market development. We have included a section on the methodology section (6) and adjusted the last section (8) to include the Salient points, conclusion and recommendations section.

Moreover, we referred to the consultation of ESMA on technical standards on trade repositories under Regulation (EU) 2019/834 (EMIR REFIT) and its’ expected implication in section (8).

2) “Ultimately, we demonstrate that the new EU regulation has provided for an effective and efficient Securitization market by ensuring that investors are protected by strict compliance to the STS Securitization criteria and proper overall supervision”: I could not find any ‘demonstration’ (i.e. at least reasoning, comparison with US, data, etc.);

This statement was changed (since this paragraph was no longer relevant) to reflect the narrowed aim and the main findings of the securitisation regulation development in Malta.

3) the reference to the NCA issue mentioned in the abstract is not discussed adequately in the text. I therefore suggest to declare in the abstract the merely descriptive nature (or, better, improve the analysis with critical aspects) and eliminate the US part; -

We eliminated the US part as this is no longer relevant to our study

  • Other aspects to be improved:
    • I would add some discussion about the need of securitizations (considering the financial crisis negative effects, why should we encourage securitizations), besides references to the EC documents –

We have added the discussion about the importance of securitization in the Introduction and in a separate new section 1.2 The importance of Securitisation.

  • Sources/references: many pillars in the literature about securitization have not been mentioned/analyzed (see Iacobucci, Kettering, De Vries Robbé, Jobst, etc.) as well as other recent papers/contributions on the topic –

We have tried to consult the most recent and what we felt as relevant sources for our study, specifically

Jobst, A, A. ‘Asset Securitisation as a risk management and funding tool,’  September 2006, Vol. 32, Issue 9,  Managerial Finance, pp.731 -760

Jobst., A, ‘What is Securitisation?,’ Finance & Development, September 2008

Bavoso, V. 2015. ‘Simple, Transparent and Standardised Securitisation: Business as usual?,’ Foundation for European Progressive Studies, September 2016

Cullen, J. 2017. ‘Securitisation, Ring - Fencing and Housing Bubbles: Financial Stability Implications of UK & EU Bank Reforms’, 2017, Research Paper No. 13, Journal of Financial Regulation

Delivorias, A. 2018. ‘Securitisation and capital requirements’, European Parliamentary Research Service, 2018

Grech, R. 2017. ‘The impact of MiFID II and MiFIR on regulated markets in Malta,’ Bachelors of  Commerce (Honours) in Banking and Finance

Panizza, R. 2018. ‘The principle of subsidiarity’, January 2018

Schwarcz, L, S. ‘Securtisation and post - crisis financial regulation,’ 2016, Vol.101, Cornell Law Review, pp.115-139

Schwarz, L, S. 2011. ‘Protecting Investors in Securitisation Transactions: Does Dodd - Frank Help or Hurt?,’ The 2011 Diane Sanger Memorial Lecture

Segoviano, M., Jones, B., Lindner, P. and Blankenheim, J. 2015. ‘Securitisation:The Road Ahead,’ January 2015, IMF Discussion Paper, pp. 1-35

 Zammit, M.2010. ‘The Securitisation process and the role of SPVs in the local financial markets,’ Masters of Arts in Financial Services, (University of Malta, 2010)

  • Minor changes required:
    • In the introduction, report data about volumes before and after the crisis (EU reforms also aim to foster the market again)

We have added some data in the introduction.

  • 2 “It is our opinion that the EU Securitization legislation is now, so far working efficiently with this Securitization regulation, even though this regulation d the cost of compliance both for the NCA's and the counter-parties”: sentence unclear/with errors –

Sentence amended as suggested.

  • Paragraph 2: need to mention the originate-to-distribute problem
  • 3, § 2.1: It seems that you are saying that the IRB method was introduced with Basel III:

We adjusted the sentence, included a reference to Basel II and outlined that Basel III retained the IRB method.

  • 8 first paragraph: put the following sentence in a footnote instead of in a parenthesis, please: “European delegated regulation together with a significant number of regulatory and implementing technical standards (In 2013, the EBA published the Regulatory Technical Standards (RTS) on Securitization retention rules and the respective requirement. Whilst the Implementing Technical Standards (ITS) relates to the convergence of supervisory practices for implementing the additional risk weights in any non-compliance retention rules”.

We inserted the sentence in a footnote as suggested.

Round 2

Reviewer 1 Report

Accept in present form.

Author Response

Thank you for your comments and suggestions which we appreciate and have taken on board. These have made our paper much better.

We have made improvements to the Methodology section and the conclusions section taking on board all your suggestions.

Thank you again for your time and for accepting publication

Reviewer 2 Report

Need to add references for §1.2 (otherwise, unsupported/lacking academic sources and fundamental research)

still minor misspellings: see §6.3 'jursidictaions'

The analysis of Malta securitization legislation (Chapter 484) is too superficial for a legal article and cannot be based on experts' opinion only (you are supposed to be the expert): you need to analyze the text of the legislation (https://legislation.mt/eli/cap/484/eng/pdf) and only then enrich it with market/supervisors' opinions, underline differences, discretions, pros/cons, etc. compared with EU law. 

Author Response

Thank you for your comments and suggestions which we appreciate and have taken on board. These have made our paper much better.

With reference to your comments and suggestions we agree and have made improvements to the paper. These have been track changed and also we have answered to the comments in bold below.

  1. The Introduction needs to provide more relevant references -Need to add references for §1.2 (otherwise, unsupported/lacking academic sources and fundamental research)

we agree and have added some references which include:

 (Orkun, 2013) Orkun Akseli. 2013. ‘Securitisation, the Financial Crisis, and the Need for Effective Risk Retention’, March 2013.

 (Grima, 2012) Grima, S., 2012. The Current Financial Crisis and Derivative Misuse, Journal of Social Sciences Research, vol. 1, no. 8, pp. 265-276.

 (European Commission, 2015a) European Commission. 2015a. ‘Fact Sheet: A European Framework for Simple and Transparent Securitisation’, Brussels,2015, available from: <http://europa.eu/rapid/press-release_MEMO-15-5733_en.htm> (last viewed on 10th November 2017)

(Deloitte, 2018) Deloitte. 2018. Securitization. Structured Finance Solutions. March 2018. Available from < https://www2.deloitte.com/content/dam/Deloitte/lu/Documents/financial-services/lu_securitization-finance-solutions.pdf> ( Last viewed on the 10th September, 2020).

 (Baums, 1994) Baums, T., 1994, Asset Securitization in Europe. Kluwer Law and Taxation Publishers; Curtin, E. & Tanega, J., 2009, Securitisation Law: EU and US Disclosure Regulations. LexisNexis. Ramos-Munoz, D. & Ingram, K., 2010, The Law of Transnational Securitization. Oxford.

 (Messina et al., 2019) Messina P. And Horrocks, M. 2019. STS Securitisation: New dispositions for a uniform European discipline.  Open Review of Management, Banking and Finance. Regents University of London Centre of Banking and Finance.

 (Global Legal Books, 2018) Global Legal Book, Securitisation 2018 (England and Wales): A Practical Cross-Border Insight into Securitization Work.

  1. still minor misspellings: see §6.3 'jursidictaions'

Agreed. The spelling has been adjusted and we have proofread the article to ensure there are no other mis-spelling or grammatical mistake

  1. The Introduction, methodology, design and conclusion needs to be described better

Agreed, we have improved this section as per your suggestions and tracked changed the changes in the article

  1. Conclusions need to be supported by results: The analysis of Malta securitization legislation (Chapter 484) is too superficial for a legal article and cannot be based on experts' opinion only (you are supposed to be the expert): you need to analyze the text of the legislation (https://legislation.mt/eli/cap/484/eng/pdf) and only then enrich it with market/supervisors' opinions, underline differences, discretions, pros/cons, etc. compared with EU law. 

         Agreed. This has been improved as per your suggestions as evidenced by the track changes in the article